# Validating Knee Varus Alignment Measurements Using Markerless Motion Capture

**DOI:** 10.3390/geriatrics8060109

**Published:** 2023-11-03

**Authors:** Kensuke Todoriki, Yoshihiro Kai, Shogo Mukai, Shin Murata

**Affiliations:** 1Graduate School of Health Sciences, Kyoto Tachibana University, Kyoto 607-8175, Japan; 2Department of Physical Therapy, Faculty of Health Sciences, Kyoto Tachibana University, Kyoto 607-8175, Japan; kai-y@tachibana-u.ac.jp (Y.K.); murata-s@tachibana-u.ac.jp (S.M.); 3Department of Orthopedic Surgery, National Hospital Organization Kyoto Medical Center, Kyoto 612-0861, Japan; shomu67d@gmail.com

**Keywords:** markerless motion capture, femorotibial angle, validity

## Abstract

This study aimed to determine the validity of specific knee varus alignment measurement methods. We measured the femorotibial angle (FTA) using radiography and optical motion capture and validated the FTA measurement using markerless motion capture. The subjects included 34 legs of 19 patients with knee osteoarthritis (OA). One-way analysis of variance and multiple comparison tests were used to compare the FTA values between the Kellgren–Lawrence classification (KL) and Pearson’s correlation coefficient for validity. The analysis showed that the FTA measured by markerless motion capture had a significant correlation to the FTA measured by radiography (r = 0.869, *p* < 0.01) and significantly increased with increasing KL (*p* < 0.05). These results indicate that markerless motion capture is a valid outcome measure for varus alignment in patients with knee OA.

## 1. Introduction

Knee osteoarthritis (OA) is a degenerative disease of knee joint components, such as articular cartilage and ligaments. It is associated with age-related degeneration and is one of the most common forms of osteoarthritis [1]. In a survey of 2213 adult men and women in the United States, knee OA was detected on radiographs in 37% of patients aged over 60 years [2]. Knee OA leads to pain [3] and a decline in physical function [4], which reduces Quality of Life (QOL) [5]. Moreover, decreased activity due to knee OA leads to obesity [6] and increases the risk of complications, such as diabetes [7] and cardiovascular and cerebrovascular diseases [8,9]. Bedson et al. [10] reported that approximately 50% of patients over the age of 50 years with knee OA, despite having difficulties in their daily lives due to pain, do not seek medical attention. This also highlights the need for prevention and early detection.

In knee OA, varus misalignment is often observed as a characteristic of lower extremity defects in the frontal plane [11]. Varus knee misalignment increases the risk of developing [12] and progressing [13] knee OA. However, it is also a factor that allows for biomechanical interventions such as knee braces [14] and plantar plates [15]. Its evaluation is essential to prevent the development and progression of knee OA. Knee varus alignment was measured as the angle between the long axes of the femur and tibia (femorotibial angle (FTA)) on a simple radiograph of the full length of the standing lower limb [16]. However, radiographic evaluation of the FTA requires sophisticated imaging techniques and involves the risk of radiation exposure due to an increased radiation dose [17]. Furthermore, a major problem is that radiography is not suitable for the prevention or early detection of knee OA outside of the hospital setting. Therefore, an evaluation method that can easily measure the FTA is desirable.

In a previous study, Stief et al. [18] investigated the relationship between FTA measured by radiography and optical motion capture (mocap method). The results showed high validity and suggested that it could be an alternative to radiography in clinical situations. However, because the subjects were healthy adolescents, whether it can be used in patients with knee OA has not been verified, and there is room for debate. In addition, markerless motion capture (MMC method), which has attracted attention in recent years, can measure the joint angles from captured images and videos through posture estimation using machine learning [19]. This method requires no special measurement equipment or environment, is easy to prepare, and can estimate the joint angles in a short time. However, the validity of the FTA measured using the MMC method versus that measured using radiography (R method) has not been verified.

Based on the above, we hypothesized that FTAs measured using the MMC and R methods are related and increase with the severity of knee OA. This study aimed to examine the validity of measuring the FTA in patients with medial knee OA using the R and MMC methods.

## 2. Materials and Methods

### 2.1. Study Design

An observational cross-sectional study was conducted on patients with knee OA admitted to Kyoto medical center for surgical treatment.

### 2.2. Subject

This study included 34 legs of 19 patients who were diagnosed with medial knee OA at Hospital A and required surgical treatment. Of the 19 subjects, four had unilateral OA, and 15 had bilateral OA, whereas 34 legs were found to have OA. The subjects were 6 males and 13 females. The mean age of the subjects was 73.7 ± 8.2 years, the mean height was 153.8 ± 11.0 cm, and the mean weight was 62.1 ± 13.8 kg. The subjects were classified as Grade II (9 limbs), Grade III (15 limbs), and Grade IV (10 limbs) according to the Kellgren–Lawrence (KL) classification. Participants were included if they agreed to participate in the study, able to maintain a stationary standing position and move indoors without aid and had no limitation of knee joint extension. The exclusion criteria were concomitant osteoarthritis of the hip or foot, a history of knee surgery, central diseases such as post-stroke syndrome or Parkinson’s disease, rheumatoid arthritis, a history of traumatic diseases such as fractures, and cognitive dysfunction. This study was conducted after an oral explanation was given, and written consent was obtained from all participants. The study was performed with ethical consideration, as stated in the Declaration of Helsinki. This study was approved by the research ethics committee of the authors’ university (approval number: 22–40).

### 2.3. FTA Measurement and Analysis Methods

FTA was measured preoperatively using three different methods: the R method, mocap method, and MMC method.

The R method was used in a stationary standing position with both feet shoulder-width apart and the tip of the foot facing forward. Radiographs focused on the knee joint, and the entire length of the lower extremity was photographed in the frontal plane. The knee joint was instructed to be fully extended and equally loaded on both legs. The R method is defined as the lateral angle consisting of the femoral and tibial axes on the frontal plane [20] (Figure 1A). The femoral and tibial axes were identified visually, and the FTA was calculated after drawing a straight line. The R method was assessed by a doctor who had been engaged in knee joint practice in the field of orthopedic surgery for more than 10 years.

The MMC method estimated the feature points of each joint from images of the whole body taken with a high-resolution camera (C992n PRO, Logicool Co Ltd., Tokyo, Japan) using Pose-Cap (4Assist Co Ltd., Tokyo, Japan), a markerless skeletal detection software. Pose-Cap is a system that incorporates the posture-estimation AI engine VisionPose (Next-System Co. Ltd., Fukuoka, Japan) and can automatically detect 30 locations (25 joints and 5 face parts) without using markers or depth sensors [19]. The MMC method was defined as a lateral angle, consisting of three points at the center of the hip (hip joint center), knee (knee joint center), and foot (ankle joint center) (Figure 1B). The limb measurement stance for the mocap and MMC methods was the same as that for the R method, with both feet shoulder-width apart, and the tip of the foot facing forward.

The mocap method was performed using an optical 3D motion capture system (OptiTrack Duo; Acuity Inc., Tokyo, Japan). Retroreflective markers were placed on the bilateral superior anterior iliac spine, knee joint center, and ankle joint center following a previous study by Yang et al. [21]. The knee and ankle joint centers were measured using calipers and defined as the midpoints of the medial and lateral clefts of the knee joint and the midpoints of the medial and lateral malleoli, respectively. The mocap method was defined as the lateral angle consisting of three points: the superior anterior iliac spine, midpoints of the medial and lateral clefts of the knee joint, and midpoints of the medial and lateral malleoli.

### 2.4. Statistical Analysis

Statistical analyses were performed using repeated measures of analysis of variance and Bonferroni’s multiple comparison test to compare the R method with the mocap and MMC methods. Next, the association between the R, mocap, and MMC methods was examined using Pearson’s correlation coefficient. A single regression analysis was also performed using the R method as the dependent variable and the mocap and MMC methods as the independent variables. One-way analysis of variance and multiple comparison tests were used to compare the FTA among the KL classifications for each method. Multiple comparisons were made using Bonferroni-corrected paired *t*-tests. All statistical analyses were performed using Microsoft windows SPSS Ver. 28.0 (IBM Japan Ltd., Tokyo, Japan) at a significance level of 5%. Moreover, the level of statistical significance for multiple comparisons was set at *p* < 0.05/3.

## 3. Results

The R method, mocap method, and MMC method averaged 183.4 ± 4.8°, 183.3 ± 5.1°, and 187.1 ± 4.6°, respectively, for the 19 subjects (Table 1). Repeated analysis of variance showed a significant difference between the FTA for each measurement method (F = 49.35, *p* < 0.01). The results of Bonferroni’s multiple comparison test showed that the MMC method was significantly better than the R and mocap methods (*p* < 0.01).

The results of the correlation analysis showed a significant correlation between the R method and mocap method (r = 0.920, *p* < 0.01) and the R method and MMC method (r = 0.869, *p* < 0.01). Furthermore, the regression equations obtained from the single regression analysis were R method = 0.861 × mocap method + 25.630 (R^2^ = 0.846) for the mocap method (Figure 2), and R method = 0.905 × MMC method + 14.153 (R^2^ = 0.755) for the MMC method (Figure 3), both of which were significant (*p* < 0.01).

A comparison of the FTA between the KL classification grades for each measurement method showed significant differences (R method: F = 23.11, *p* < 0.01; mocap method: F = 15.59, *p* < 0.01; MMC method: F = 12.47, *p* < 0.01) (Figure 4). The results of the Bonferroni’s multiple comparison test showed that the FTA measurements obtained using each method increased significantly with increasing grades (*p* < 0.01, *p* < 0.05).

## 4. Discussion

This study investigated the validity of the mocap and MMC methods for measuring the FTA in patients with knee OA. The results showed a significantly strong positive correlation between FTA values from the R method and mocap and MMC methods. Moreover, the FTA measured using each of the three methods increased with knee OA severity. Therefore, the MMC method is a valid outcome measure of varus alignment in patients with knee OA.

The measurements obtained in this study based on the R method were more approximate compared to previous studies [22]. This confirmed that the FTA increased with the severity of knee OA in the patients included in this study.

The MMC method exhibited a significant correlation with the R method. Furthermore, the results of a single regression, using the R method as the dependent variable and the MMC method as the independent variable, showed that the regression equation obtained was significant. Previous studies have also examined non-radiographic methods for assessing knee varus alignment. Kraus et al. [23] proposed a clinical approach to quantify the alignment of the frontal plane by measuring the angle between the femoral and inferior femoral axes from the body surface using a goniometer. However, this method tends to underestimate knee varus alignment [24] and has poor validity [25]. Ohnishi et al. [26] reported an association between a template-matching method to measure knee varus alignment and the FTA from radiographs. However, this method is less convenient because it requires a special imaging device and a complicated processing procedure to calculate the measured values. However, the MMC method used in this study does not require marker attachment and allows for in situ calculation of the measured values. In this study, the MMC method showed a strong positive correlation with the R method, which is the gold standard for measuring knee varus alignment. This suggests that the MMC method is valid as a simple method for evaluating knee varus alignment.

However, the angle obtained from the MMC method was significantly larger than that of the R and mocap methods by an average of approximately 5°. This difference is thought to be due to the reference points for the R and mocap methods being outside the center of the hip, whereas the MMC method is based on the center of the hip. In a previous study examining the relationship between the mocap and MMC methods in healthy adult males, it was reported that the angle obtained from the MMC method was significantly higher, with a mean difference of approximately 5° [27]. Therefore, the MMC method may overestimate by approximately 5° compared to the R and mocap methods.

A comparison of the FTA between the KL classification grades by each measurement method showed that FTA increased significantly with increasing severity of knee OA for all measurement methods. Previous studies [28,29] have reported an association between the progression of knee OA severity and increased FTA in patients with knee OA. However, the method of measuring varus knee alignment [23] that has been used until now is not suitable for understanding changes in severe internal knee misalignment owing to inaccuracies [30].

This study also has several limitations. First, the R method, mocap method, and MMC method could not be used simultaneously. Therefore, the measurement posture was defined, and measurements were performed to reduce posture errors. Second, as the study included patients with knee OA who required surgical treatment, the severity of OA was skewed, and the sample size for each grade was small. In the future, it will be necessary to increase the number of participants in each grade to verify these details.

## 5. Conclusions

In this study, the FTA was measured using radiography and optical motion capture, and the measurement was validated using markerless motion capture. The results showed that the R and MMC methods were significantly related to each other. In addition, the FTAs measured using both methods increased significantly with the progression of knee OA. This result suggests that the MMC method can adequately assess the severity of the deformity.

## Figures and Tables

**Figure 1 geriatrics-08-00109-f001:**
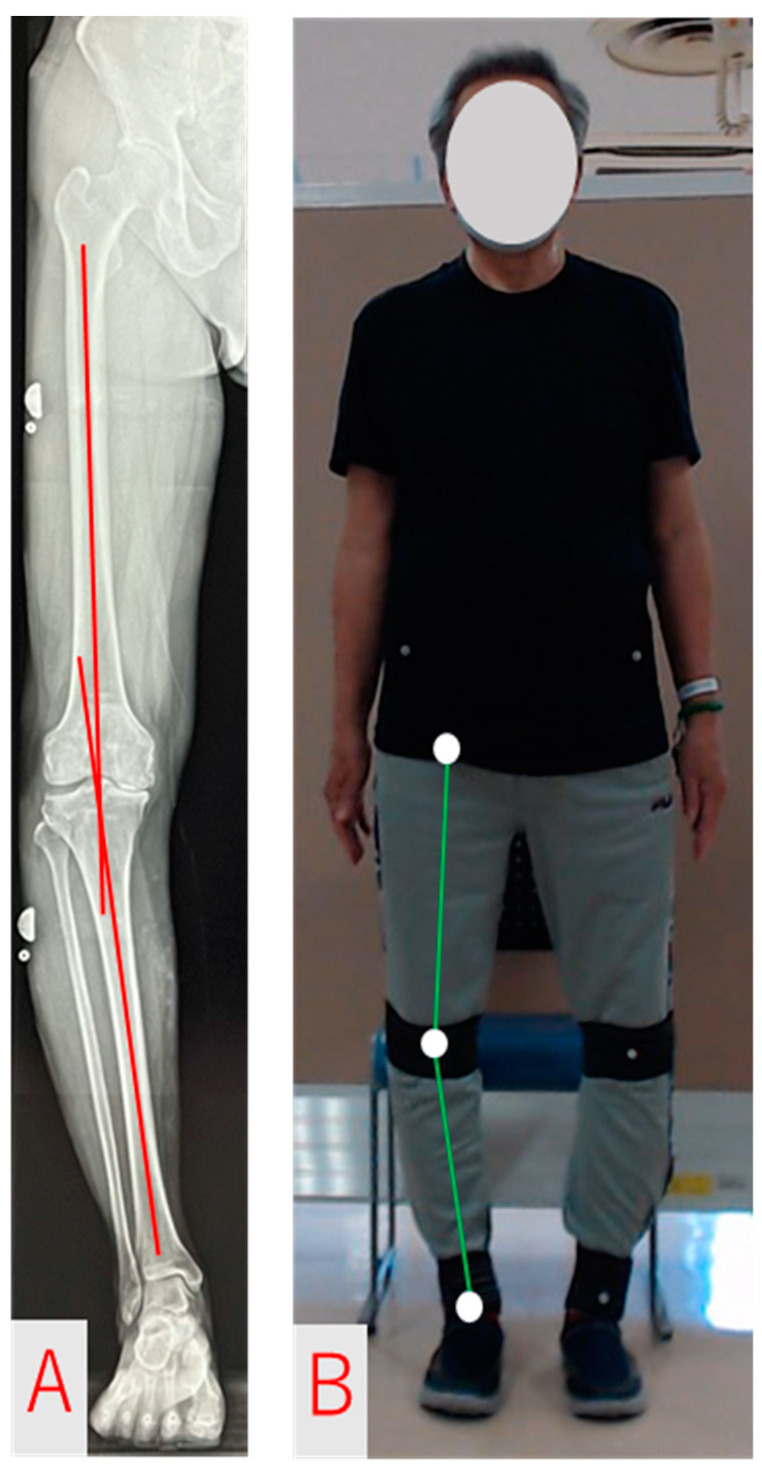
Images for femoro-tibial angle measurement. (**A**) R method. (**B**) MMC method.

**Figure 2 geriatrics-08-00109-f002:**
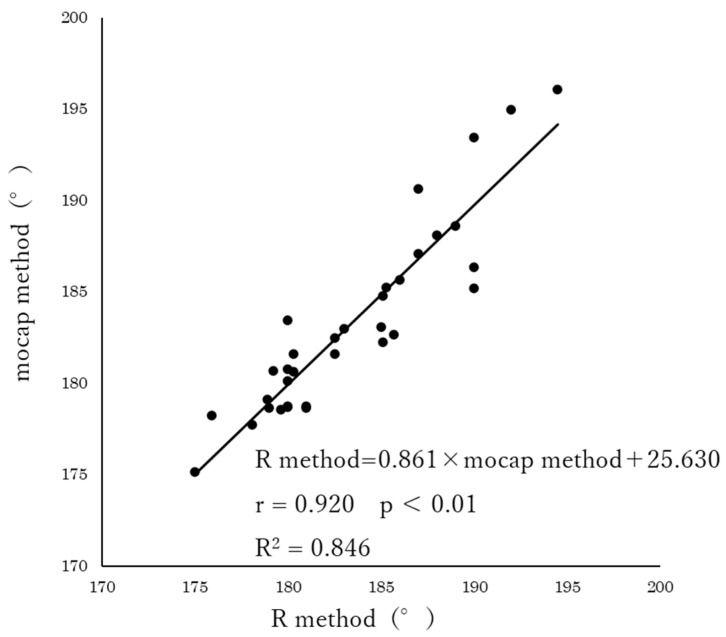
Relationship between R method and mocap method (*n* = 34).

**Figure 3 geriatrics-08-00109-f003:**
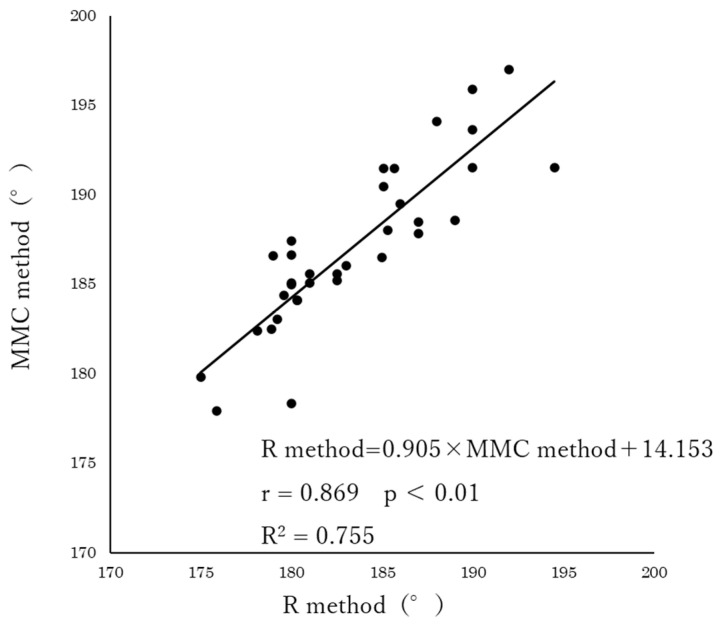
Relationship between R method and MMC method (*n* = 34).

**Figure 4 geriatrics-08-00109-f004:**
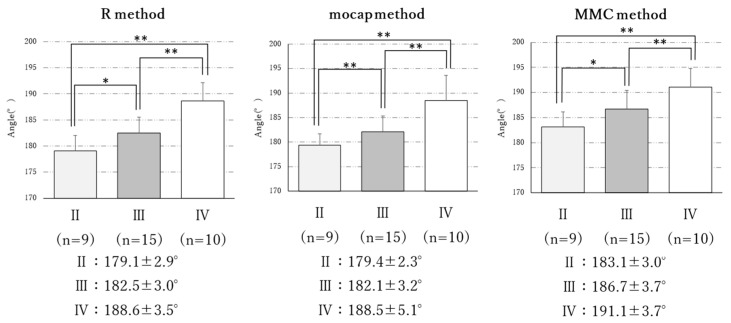
Comparison of FTA between Kellgren–Lawrence classifications by each measurement method (*n* = 34). Mean ± SD. *: *p* < 0.05/3, **: *p* < 0.01.

**Table 1 geriatrics-08-00109-t001:** Comparison of R method with mocap method and MMC method measurement (*n* = 34).

	R Method (A)	Mocap Method (B)	MMC Method (C)	*p*-Value	Multiple Comparison
Measurements	183.4 ± 4.8	183.3 ± 5.1	187.1 ± 4.6	<0.01	A, B < C **

**: *p* < 0.01.

## Data Availability

The data used to support the findings of this study are available from the corresponding author upon request. The data are not publicly available because they contain information that can compromise the privacy of research participants.

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
