# Peer review of "Validating Knee Varus Alignment Measurements Using Markerless Motion Capture"

_geriatrics, 2023, doi:10.3390/geriatrics8060109_

Round 1

Reviewer 1 Report

Comments and Suggestions for Authors

This report presented the validation of markerless motion capture for FTA measurement as compared to standing leg radiograph and optical motion capture.  MMC is convenient  and no radiation. This is useful in w/o hospital setting. 

There are several points to clear in this manuscript.

1. FTA measurement

In radiology, the patella of the interest knee is in front of the x-ray. How about MMC method? In the measurement of MMC, the knee may be externally rotated especially in KL4. And also, the knee could not be full extension in KL4. There may be a larger measurement error. And femoral  or tibial bone rotation is also striking in KL4

.

Comments on the Quality of English Language

1.P3,L107&110

lateral cleft:   may be lateral joint line?

2.Need references abbreviation

#11, #15, #26

Reviewer 2 Report

Comments and Suggestions for Authors

The authors have presented a manuscript that compared three methods (radiograph, motion capture and markerless motion capture) for quantifying standing femorotibial angle (or varus / valgus alignment) for individuals with documented knee OA (KL II to IV). They found that the markerless motion capture FTA measures were significantly different (greater) from the radiograph and motion capture measures; but all methods were significantly correlated and quantified a significant difference in FTA between KL groups. This reviewer thinks the authors have presented a nice, well justified study. I thought the clearly concise introduction to a real strength of the presented work. I am unsure about the audience of the chosen journal, but I do believe this work would be of interest to a certain audience. However, I have several concerns about the presentation of the current experimental findings, in particular, the interpretation of the findings.

First and foremost, this reviewer does not believe the findings demonstrate the markerless motion capture method to be valid for determining FTA, as they were significantly different than the current gold standard. Next, I believe the authors need to compare their FTA values for all methods with existing experimental evidence in the discussion, and interpret their similarity or differences. Finally, the work lacked conclusion statements, and provide a barebones limitations and suggestions for future work in the conclusion section.

Also, this reviwer strongly suggests authors avoid using acronyms. Using R and MMC required an unnecessary cognitive load, and ultimately detracted from my comprehension.

Intro:

I thought the introduction was clear, concise, and the strength of the current manuscript. However, I believe providing a directional hypothesis would be very useful for the reader, and strengthen the introduction.

Methods:
The methods were clear and could be replicated. However, I do believe the authors need to bolster the explanation for quantifying FTA from the radiograph and revise the statistical analysis section (it was wordy and a little hard to follow).

Line 121: typo: Microsoft window SPSS

Results:

This reviewer believes the authors could revise (be much more concise) with results presentation, and improve the reader comprehension. It was wordy.

Lines 125-127: The ANOVA identified a main effect of measurement method (R, MMC or mocap) on FTA, and did not show a difference between all three. In fact, the post hoc analysis (lines 127-128) identified MMC significantly differed from R and mocap, but although unspecified in the text, there was no significant difference between R and mocap methods. Additionally, they statistical results do not demonstrate or determine whether any method is better than another method (line 128).

Lines 131-133: What about the correlation between MMC and mocap derived measures?

Lines 133-136: Personally, this reviewer does not think you need to report the regression equations, but rather the R^2 values, or degree of agreement between the two methods.

Lines 141-145: This reviewer is a little confused on the results presented here. I believe the authors ran a one-way ANOVA to determine whether each method (R, MMC and mocap) identified significant differences in FTA between KL groups. Is that true? If so, the results could be presented more succinctly. Also, were there not significant differences between KL II and IV for each method? In addition, did the authors compare FTA for each KL group between each method (R, MMC and mocap)? If not, why? Is that not useful for information?

Discussion:

This was a clear weakness, and requires substantial work in order to make this a publishable manuscript.  

Lines 152: Did not you not test the validity of using mocap and MMC to measure FTA, as R would be considered the “gold” standard?

Lines 156-157: This reviewer does not believe the authors, based on the current experimental evidence, can make this statement. The MMC method significantly differed from the R (the gold standard) as well as mocap derived measures. I believe the MMC detected differences between KL groups, but may overestimate FTA.

Yes (lines 158-161), as R derived FTA increased so did MMC derived MMC, but it appears at each point that the MMC values were larger the R derived values. Sort of address this below.

This reviewer believes the authors need to report and discuss the substantial difference in FTA measured between R and MMC. This is a critical limitation. In addition, how do your FTA values compare to previously published data? Are they similar?

Lines 183: But, you have just spent the first half the discussion telling the reader it is a valid measure.

Limitations and future work – why in the conclusion section? There was no conclusion statements.  

Comments on the Quality of English Language

I have not issues with the english. There were a few minor typos and grammatical errors, that commonly occur in any paper. 

Reviewer 3 Report

Comments and Suggestions for Authors

The authors conducted a study to investigate “Validating knee varus alignment measurements using marker-less motion capture”. The study is a interesting study. However, some concerned were found in your study. Accordingly, this reviewer would like to suggest to the authors base on the following comments.

This study is a study to confirm the feasibility of using the MMC method compared to the R method and the mocap method.

However, the angle is about 4 degrees larger than the R method and the Mocap method. This can be a factor that can change the KL grade, making accurate evaluation difficult.

As such, I disagree with the idea that validity is judged by statistical significance for the three methods.

In the statistical method, the Bonferroni method should use the adjusted p value. Comparative values between groups should be presented.

Round 2

Reviewer 1 Report

Comments and Suggestions for Authors

This manuscript is suitable for Geriatrics.